# RelationMatch: Matching In-batch Relationships for Semi-supervised Learning

## Abstract

Semi-supervised learning has gained prominence for its ability to utilize limited labeled data alongside abundant unlabeled data. However, prevailing algorithms often neglect the relationships among data points within a batch, focusing instead on augmentations from identical sources. In this work, we introduce RelationMatch, an innovative semi-supervised learning framework that capitalizes on these relationships through a novel Matrix Cross-Entropy (MCE) loss function. We rigorously derive MCE from both matrix analysis and information geometry perspectives. Our extensive empirical evaluations, including a 15.21% accuracy improvement over FlexMatch on the STL-10 dataset, demonstrate that RelationMatch consistently outperforms existing state-of-the-art methods.

## 1 Introduction

Semi-supervised learning lives at the intersection of supervised learning and self-supervised learning (Tian et al., 2020; Chen et al., 2020a), as it has access to a small set of labeled data and a huge set of unlabeled data. In order to fully harness the potential of these two data types, techniques from both realms are employed: it fits the labels using the labeled data and propagates the labels on the unlabeled data with prior knowledge on the data manifold. With this idea, semi-supervised learning has achieved outstanding performance with very few labeled data, compared with the supervised learning counterparts (Sohn et al., 2020; Zhang et al., 2021; Wang et al., 2022c).

The state-of-the-art semi-supervised learning algorithms are mostly based on a notion called pseudo label (Lee et al., 2013; Tschannen et al., 2019; Berthelot et al., 2019b; Xie et al., 2020; Sohn et al., 2020; Gong et al., 2021), generated on the fly for the unlabeled data by the neural network $f$ during training. Such ideas can be traced back to self-training in Yarowsky (1995). Specifically, in each iteration, both labeled and unlabeled data points are sampled. For the unlabeled data points, weak augmentations are applied, followed by evaluating the confidence of network $f$ in labeling these inputs. If high confidence is established, the predicted labels are recognized as pseudo labels for the unlabeled data points. We subsequently train $f$ to predict the same label for their strongly augmented counterparts.

Essentially, two key steps facilitate the exploitation of the unlabeled dataset. First, if $f$ exhibits confidence in the weakly augmented data point, we record the prediction as pseudo labels. Secondly, we expect that $f$ upholds consistency between weak and strong augmentations for each (pseudo) labeled data point, based on the prior that they convey the same (albeit possibly distorted) semantic meaning. For instance, given an image $\mathbf{x}$ and its weak/strong augmentations $\mathbf{x}^w, \mathbf{x}^s$, if $f$ asserts $\mathbf{x}^w$ to be a cat with high probability, then $f$ should also recognize $\mathbf{x}^s$ as a cat, not a different animal. However, is the consistency between each pair of weak and strong augmentations the only information that we can use for semi-supervised learning?

In this paper, we propose to additionally enforce the consistency between the in-batch relationships of weak/strong augmentations in each batch. See Figure 1. In the upper row, the four images are only weakly augmented, and assume that $f$ gives the correct pseudo-labels for them. For the strongly augmented images in the lower row, the existing algorithms only consider pairwise consistency using cross-entropy, which means the prediction of the strongly augmented dog shall be close to the one-hot vector for dog, and the strongly augmented cat shall be close to the one-hot vector for cat, etc. In addition to this regularization, we propose RelationMatch, which uses the matrix cross-entropy loss (MCE) to capture the in-batch relationships between the images. Therefore, we hope $f$ believes the

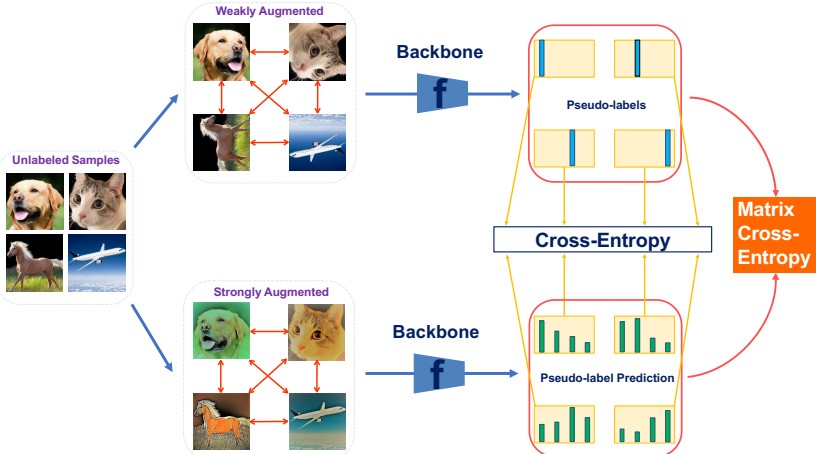

Figure 1: Pseudo-labels are obtained by feeding a batch of weakly-augmented images into the model. The model then predicts probabilities for strongly-augmented versions of these images. The loss function incorporates both cross-entropy and matrix cross-entropy loss.

relationship between the weakly augmented dog and weakly augmented cat, shall be close to the relationship between the strongly augmented dog and strongly augmented cat:

$$\mathrm{Relation}(\text{WeaklyAug dog}, \text{WeaklyAug cat}) \approx \mathrm{Relation}(\text{StronglyAug dog}, \text{StronglyAug cat}).$$

Formally, we represent each image $\mathbf{x}$ with the prediction vector $\boldsymbol{f}(\mathbf{x}) \in \mathbb{R}^k$. We use the inner products between images to represent their relationship. Notice that such relationships are always computed for the same type of augmentations, e.g., between weakly augmented dog and weakly augmented cat, but never between weakly augmented dog and strongly augmented cat. For each mini-batch of samples of $b$ images, consider their weak (or strong) augmentations. Using $\boldsymbol{f}$ to represent these images, we get a relationship matrix $\mathbf{A} \in \mathbb{R}^{b \times k}$, where each row of $\mathbf{A}$ represents the prediction vector of an image. By computing $\mathrm{R}(\mathbf{A}) \triangleq \mathbf{A}\mathbf{A}^{\top}$, we get a $b \times b$ matrix, which stores all the relationships between any two images in the batch. Notice that we will compute different relationship matrices for weak/strong augmentations, denoted as $\mathrm{R}(\mathbf{A}^w)$ and $\mathrm{R}(\mathbf{A}^s)$, respectively.

To define the loss for matching $\mathrm{R}(\mathbf{A}^w)$ and $\mathrm{R}(\mathbf{A}^s)$, we adopt two distinct theoretical perspectives to generalize the cross-entropy loss of vectors to MCE, deriving from both matrix analysis and information geometry. Intriguingly, our MCE loss emerges as the natural choice from both aspects and possesses numerous desirable properties. The diagram of RelationMatch is presented in Figure 1. In our experiments, we observe RelationMatch incurs a significant performance uplift for STL-10, and consistent improvements for CIFAR-10 and CIFAR-100 as well. Interestingly, we find that it also proves effective for supervised learning scenarios.

Our contributions can be summarized in three folds:

- The introduction of RelationMatch, a novel SSL algorithm that captures in-batch relationship consistency.

- The development of the MCE loss function, underpinned by two separate theoretical frameworks, which exhibits a number of desirable properties including convexity, lower bounded, and minimization properties.

- Extensive empirical validation on vision benchmarks such as CIFAR-10, CIFAR-100, and STL-10, where RelationMatch consistently outperforms state-of-the-art methods. Remarkably, on the STL-10 dataset with only 40 labels, our method outperforms the well-known FlexMatch (Zhang et al., 2021) by 15.21%. It also has consistent improvements for supervised learning scenarios.

## 2 MATRIX CROSS-ENTROPY FOR SUPERVISED AND SEMI-SUPERVISED LEARNING

### 2.1 WARM-UP EXAMPLE

To elucidate how our algorithm captures relationships through both weak and strong augmentations, let's begin with a straightforward example. Suppose we have $b = 4, k = 3$, where three out of the four images belong to the first class, and the fourth image belongs to the last class. We assume that the function $f$ assigns accurate pseudo-labels for the weak augmentations, denoting $\mathrm{R}(\mathbf{A}^w) = \mathbf{A}^w(\mathbf{A}^w)^\top$ as:

$$\mathrm{R}(\mathbf{A}^w) = \mathrm{R}\left(\begin{bmatrix} 1 & 0 & 0 \\ 0 & 0 & 1 \\ 1 & 0 & 0 \\ 1 & 0 & 0 \end{bmatrix}\right) = \begin{bmatrix} 1 & 0 & 1 & 1 \\ 0 & 1 & 0 & 0 \\ 1 & 0 & 1 & 1 \\ 1 & 0 & 1 & 1 \end{bmatrix} \tag{2.1}$$

Since the pseudo labels for weak augmentations are always one-hot vectors, $\mathrm{R}(\mathbf{A}^w)$ is well structured. Specifically, for rows that are the same in $\mathbf{A}^w$, they are also the same in $\mathrm{R}(\mathbf{A}^w)$, with values representing the corresponding row indices. In other words, $\mathrm{R}(\mathbf{A}^w)$ represents $k$ distinct clusters of one-hot vectors in the mini-batch.

If $f$ can generate exactly the same prediction matrix $\mathbf{A}^s$ for the strongly augmented images, our algorithm will not incur any additional loss compared with the previous cross-entropy based algorithms. However, $\mathbf{A}^s$ and $\mathbf{A}^w$ are generally different, where our algorithm becomes useful. For example, given a pair of prediction vectors $(p, q)$, if we know $p = (1, 0, 0)$, then cross-entropy loss is simply $p_1 \log q_1 = \log q_1$. Therefore, we will get the same loss for $q = (0.5, 0.5, 0)$, $q = (0.5, 0.25, 0.25)$, or $q = (0.5, 0, 0.5)$. Consider the following two possible cases of $\mathrm{R}(\mathbf{A}^s)$ generated by $f$ during training:

$$\mathrm{R}\left(\begin{bmatrix} 0.5 & 0.5 & 0 \\ 0 & 0 & 1 \\ 0.5 & 0.5 & 0 \\ 0.5 & 0.5 & 0 \end{bmatrix}\right) = \begin{bmatrix} 0.5 & 0 & 0.5 & 0.5 \\ 0 & 1 & 0 & 0 \\ 0.5 & 0 & 0.5 & 0.5 \\ 0.5 & 0 & 0.5 & 0.5 \end{bmatrix}$$

$$\mathrm{R}\left(\begin{bmatrix} 0.5 & 0.5 & 0 \\ 0 & 0 & 1 \\ 0.5 & 0.25 & 0.25 \\ 0.5 & 0 & 0.5 \end{bmatrix}\right) = \begin{bmatrix} 0.5 & 0 & 0.375 & 0.25 \\ 0 & 1 & 0.25 & 0.5 \\ 0.375 & 0.25 & 0.375 & 0.375 \\ 0.25 & 0.5 & 0.375 & 0.5 \end{bmatrix}$$

If we only use cross-entropy loss, these two cases will give us the same gradient information. However, by considering the in-batch relationships, it becomes clear that these two cases are different: the first case always makes mistakes on the second class, while the second case makes relatively random mistakes. Therefore, by comparing $\mathrm{R}(\mathbf{A}^s)$ with $\mathrm{R}(\mathbf{A}^w)$ defined in Eqn. (2.1), we can get additional training signals. In our example, the first case will not give additional gradient information for the second row (and the second column due to symmetry), but the second case will.

### 2.2 MATRIX CROSS-ENTROPY

We employ the Matrix Cross-Entropy (MCE) loss to quantify the dissimilarity between $\mathrm{R}(\mathbf{A}^w)$ and $\mathrm{R}(\mathbf{A}^s)$, which are both positive semi-definite matrices. The loss is formally defined as follows:

**Definition 2.1** (Matrix Cross-Entropy for Positive Semi-Definite matrices). For positive semi-definite matrices $\mathbf{P}, \mathbf{Q}$, the Matrix Cross-Entropy (MCE) is:

$$\mathrm{MCE}(\mathbf{P}, \mathbf{Q}) = \mathrm{tr}(-\mathbf{P} \log \mathbf{Q} + \mathbf{Q}). \tag{1}$$

Here, $\mathrm{tr}$ represents the matrix trace, and $\log$ is the principal matrix logarithm (see Appendix A.1). In fact, for $l_2$ normalized representation vectors, we have the following simplified expression:

**Proposition 2.2.** If $\mathbf{P} = \frac{1}{b}\mathbf{Y}\mathbf{Y}^\top \in \mathbb{R}^{b \times b}$ and $\mathbf{Q} = \frac{1}{b}\mathbf{X}\mathbf{X}^\top \in \mathbb{R}^{b \times b}$ are batch-normalized relationship matrices, where $\mathbf{Y}$ and $\mathbf{X}$ consist of row $l_2$-normalized vectors, then

$$\mathrm{MCE}(\mathbf{P}, \mathbf{Q}) = \mathrm{tr}(-\mathbf{P} \log \mathbf{Q}) + 1. \tag{2}$$

*Proof.* Given that the vectors in $\mathbf{X}$ are $l_2$-normalized, the diagonal elements of $\mathbf{X}\mathbf{X}^\top$ are all 1. Therefore, $\text{tr}(\mathbf{X}\mathbf{X}^\top) = b$. Substituting this into the expression for $\text{MCE}(\mathbf{P}, \mathbf{Q})$, we find $\text{MCE}(\mathbf{P}, \mathbf{Q}) = \text{tr}(-\mathbf{P}\log\mathbf{Q}) + 1$. □

At the first glance, the MCE loss looks pretty complicated. However, as we will discuss in Section 4, it can be naturally derived from both matrix analysis (Section 4.1) and information geometry (Section 4.3). Moreover, it has a nice interpretation from matrix eigen-decomposition (Appendix A.2). In Section 5, we further demonstrate that the standard cross-entropy loss can be seen as a special case of MCE, as well as some of its nice properties.

## 3 RELATIONMATCH: APPLYING MCE FOR SEMI-SUPERVISED LEARNING ON VISION TASKS

We consider a general scheme (Lee et al., 2013; Gong et al., 2021) that unifies many prior semi-supervised algorithms (including FixMatch and FlexMatch):

$$\theta_{n+1} \leftarrow \arg\min_\theta \left\{ \mathcal{L}_{\text{sup}}(\theta) + \mu_u \mathcal{L}_u(\theta; \theta_n) \right\}, \tag{3}$$

where $\theta_n$ denotes the model parameters at the $n$-th iteration, $\mathcal{L}_{\text{sup}}(\theta)$ is a supervised loss. The unsupervised loss term $\mathcal{L}_u(\theta; \theta_n)$ acts as a consistency regularization (based on the $n$-th step backbone) that operates on the unlabeled data, and $\mu_u$ is a loss balancing hyperparameter.

During training, we always have access to some labeled data and unlabeled data. Assume there are $b_s$ labeled images and $b_u$ unlabeled images. For labeled data, the loss is the classical cross-entropy loss. For those $b_u'$ unlabeled data that are pseudo-labeled, we also have CE combined with MCE (where we use pseudo labels provided by weakly augmented images, denoted as $\tilde{\mathbf{Y}}_w = [\tilde{\mathbf{y}}_1, \cdots, \tilde{\mathbf{y}}_b]^\top \in \mathbb{R}^{b \times k}$, and we denote the prediction vectors of strongly augmented images as $\tilde{\mathbf{X}}_s = [\tilde{\mathbf{x}}_1, \cdots, \tilde{\mathbf{x}}_b]^\top \in \mathbb{R}^{b \times k}$). In summary, we have:

$$\begin{aligned} \mathcal{L}_{\text{RelationMatch}}(Y, X) &= \mathcal{L}_{\text{sup}}(Y_{\text{sup}}, X_{\text{sup}}) + \mu_u \mathcal{L}_u(\tilde{\mathbf{Y}}_w, \tilde{\mathbf{X}}_s) \\ &= \sum_{i=1}^{b_s} \text{CE}(y_i, x_i) + \mu_u \left( \sum_{i=1}^{b_u} \text{CE}(\tilde{y}_i, \tilde{x}_i) + \gamma_u \cdot \text{MCE}(\text{R}(\tilde{\mathbf{Y}}_w), \text{R}(\tilde{\mathbf{X}}_s)) \right), \end{aligned} \tag{4}$$

where $\text{R}(\tilde{\mathbf{Y}}_w) = \frac{1}{b}\tilde{\mathbf{Y}}_w\tilde{\mathbf{Y}}_w^\top$, $\text{R}(\tilde{\mathbf{X}}_s) = \frac{1}{b}\tilde{\mathbf{X}}_s\tilde{\mathbf{X}}_s^\top$ are batch-normalized relationship matrices.

**Note**. We remark that Section 4 and Section 5 are purely theoretical and technical. Skipping these two sections does not affect understanding and implementing our RelationMatch method. Therefore, we choose to present our experimental results first, readers interested in the theoretical aspects are encouraged to consult Sections 4 and 5 for a deeper understanding.

### 3.1 DATASET

**CIFAR-10/100.** The CIFAR-10 dataset (Krizhevsky et al., 2009) is a benchmark in image classification, consisting of 60,000 images distributed across 10 distinct classes. Each class has 5,000 images in the training set and 1,000 in the test set, all of which are $3 \times 32\times$ pixels. CIFAR-100 (Krizhevsky et al., 2009) extends this dataset to 100 classes, each containing 500 training and 100 test images.

**STL-10.** The STL-10 dataset (Coates et al., 2011) is another widely-used resource for semi-supervised learning, derived from the larger ImageNet dataset (Deng et al., 2009). STL-10 comprises 10 labeled classes, providing 500 training and 800 test images per class. Additionally, it includes 100,000 unlabeled images, some of which belong to classes outside the labeled set. All images are $3 \times 96 \times 96$ pixels.

### 3.2 EXPERIMENT DETAILS

**Implementation details.** We adopt TorchSSL (Zhang et al., 2021) as our implementation framework, which serves as the official codebase for FlexMatch (Zhang et al., 2021) and is built upon

PyTorch (Paszke et al., 2019). We extend the TorchSSL framework to compute the Matrix Cross-Entropy (MCE) loss alongside the traditional unsupervised cross-entropy loss. For a comprehensive discussion on the implementation, refer to Appendix B.

**Hyperparameters.** To ensure a fair comparison, we adopt the same hyperparameters as used in FixMatch (Sohn et al., 2020). Specifically, we set $\gamma_u = 1$ and $\mu_u = 3 \times 10^{-3}$. Optimization is performed using SGD with a momentum of 0.9 and weight decay of $5 \times 10^{-4}$. The learning rate is initialized at 0.03 and adjusted via a cosine scheduler. The training batch size is set to 64, with a 7:1 ratio of unlabeled to labeled data. We employ a threshold $\tau$, of 0.95.

**Baselines.** We consider prior semi-supervised learning methods similar to FixMatch, including Π-Model, Pseudo-Label (Lee et al., 2013), VAT (Miyato et al., 2018), MeanTeacher (Tarvainen & Valpola, 2017), MixMatch (Berthelot et al., 2019b), ReMixMatch (Berthelot et al., 2019a), UDA (Xie et al., 2020), Dash (Xu et al., 2021), MPL (Pham et al., 2021), FixMatch (Sohn et al., 2020), FlexMatch (Zhang et al., 2021). Most baseline results are directly obtained from TorchSSL. For recent and future works on improving pseudo-label quality such as CPL introduced in Flexmatch, our method can be easily incorporated with them.

### 3.3 EXPERIMENTAL RESULTS ON SUPERVISED LEARNING

We commence our evaluation with results in a fully supervised setting, utilizing WideResNet-28-2, ResNet18, and ResNet50 as backbone architectures. Training spans 200 epochs, leveraging a cosine learning rate scheduler and a batch size of 64. For CIFAR-10 and CIFAR-100, we set $\gamma_s$ (the relative ratio of MCE loss to CE loss) to 0.1 and 0.01, respectively. The results are summarized in Table 1, which highlights the robust performance gains achieved by incorporating Matrix Cross-Entropy (MCE) across different architectures and datasets. Models augmented with MCE consistently outperform those using only cross-entropy or label-smoothing. The results affirm MCE's versatility and efficacy, suggesting that it can serve as a valuable addition to existing supervised learning techniques.

Table 1: Accuracy results of our method compared to baselines under supervised settings, WRN means WideResNet, **textbfbold** means the best, underline means the second.

| Dataset | CIFAR-10 | | | CIFAR-100 | | |
|---|---|---|---|---|---|---|
| # Backbone | WRN-28-2 | ResNet18 | ResNet50 | WRN-28-2 | ResNet18 | ResNet50 |
| only cross-entropy | $94.45_{\pm0.19}$ | $95.08_{\pm0.09}$ | $95.32_{\pm0.18}$ | $76.40_{\pm0.31}$ | $78.07_{\pm0.16}$ | $79.07_{\pm0.43}$ |
| w/ label-smoothing | $94.72_{\pm0.05}$ | $95.25_{\pm0.13}$ | $95.10_{\pm0.32}$ | $76.81_{\pm0.18}$ | **$78.41_{\pm0.21}$** | $78.70_{\pm0.44}$ |
| w/ matrix cross-entropy | **$94.79_{\pm0.05}$** | **$95.31_{\pm0.08}$** | **$95.46_{\pm0.16}$** | **$76.92_{\pm0.17}$** | $78.37_{\pm0.14}$ | **$79.11_{\pm0.52}$** |

While label smoothing (Szegedy et al., 2016) has been an effective technique for enhancing generalization in various tasks such as image classification and language translation, its application has been questioned in contexts like knowledge distillation, where it's argued to potentially erase valuable information (Müller et al., 2019). In contrast, our Lemma 3.1 establishes that Matrix Cross-Entropy (MCE) retains the one-hot properties of the target distribution without information loss.

**Lemma 3.1** (One-hot Property Preservation). *Let $\mathbf{Z}_1 \in \mathbb{R}^{b \times k}$ represent one-hot encoded probabilities of a batch of images, and $\mathbf{Z}_2 \in \mathbb{R}^{b \times k}$ be their predicted probabilities. If $\mathbf{Z}_1\mathbf{Z}_1^\top = \mathbf{Z}_2\mathbf{Z}_2^\top$, then each row of $\mathbf{Z}_2$ will also be one-hot, ensuring class consistency between $\mathbf{Z}_1$ and $\mathbf{Z}_2$.*

*Proof.* Note a vector lying on probability has its $l_2$ norm equal to 1 iff it is one hot. By analyzing each diagonal entry of $\mathbf{Z}_2\mathbf{Z}_2^\top$, it is clear that each row of $\mathbf{Z}_2$ will be one hot. The rest of the argument is clear by analyzing each off-diagonal entry of $\mathbf{Z}_2^\top\mathbf{Z}_2$. □

Lemma 3.1 elucidates that MCE captures second-order equivalences between $\mathbf{Z}_1$ and $\mathbf{Z}_2$, preserving their clustering patterns. However, this does not imply $\mathbf{Z}_1 = \mathbf{Z}_2$. For instance, consider the relationship matrices derived from different input matrices, yet resulting in identical outputs, as demonstrated below:

$$\text{R}\left(\begin{bmatrix} 0 & 0 & 1 \\ 1 & 0 & 0 \\ 0 & 0 & 1 \\ 0 & 0 & 1 \end{bmatrix}\right) = \begin{bmatrix} 1 & 0 & 1 & 1 \\ 0 & 1 & 0 & 0 \\ 1 & 0 & 1 & 1 \\ 1 & 0 & 1 & 1 \end{bmatrix}$$

This observation further emphasizes the unique ability of MCE to maintain class-consistency and clustering patterns, making it a compelling alternative to label smoothing.

## 3.4 EXPERIMENTAL RESULTS ON SEMI-SUPERVISED LEARNING

Table 2 presents our empirical evaluation of RelationMatch against benchmark models on various datasets. Specifically, the RelationMatch with Curriculum Pseudo Labeling (CPL) variant builds upon FlexMatch (Zhang et al., 2021). Across the board, RelationMatch exhibits superior performance, particularly outclassing FixMatch and FlexMatch on the STL-10 dataset when limited to only 40 labels. Importantly, the MCE loss in RelationMatch **does not interfere with the quality of pseudo-labels**, offering a modular component that can **seamlessly integrate** with future semi-supervised learning frameworks.

Table 2: Error rates (100% - accuracy) on CIFAR-10, CIFAR-100, and STL-10 dataset of state-of-the-art methods for semi-supervised learning. **bold** means the best, underline means the second.

| Dataset | CIFAR-10 | | | CIFAR-100 | | | STL-10 | | |
|---|---|---|---|---|---|---|---|---|---|
| # Label | 40 | 250 | 4000 | 400 | 2500 | 10000 | 40 | 250 | 1000 |
| Π Model (Rasmus et al., 2015a) | $74.34_{\pm1.76}$ | $46.24_{\pm1.29}$ | $13.13_{\pm0.59}$ | $86.96_{\pm0.80}$ | $58.80_{\pm0.66}$ | $36.65_{\pm0.00}$ | $74.31_{\pm0.85}$ | $55.13_{\pm1.50}$ | $32.78_{\pm0.40}$ |
| Pseudo Label (Lee et al., 2013) | $74.61_{\pm0.26}$ | $46.49_{\pm2.20}$ | $15.08_{\pm0.19}$ | $87.45_{\pm0.85}$ | $57.74_{\pm0.28}$ | $36.55_{\pm0.24}$ | $74.68_{\pm0.99}$ | $55.45_{\pm2.43}$ | $32.64_{\pm0.71}$ |
| VAT (Miyato et al., 2018) | $74.66_{\pm2.12}$ | $41.03_{\pm1.79}$ | $10.51_{\pm0.12}$ | $85.20_{\pm1.40}$ | $46.84_{\pm0.79}$ | $32.14_{\pm0.19}$ | $74.74_{\pm0.38}$ | $56.42_{\pm1.97}$ | $37.95_{\pm1.12}$ |
| MeanTeacher (Tarvainen & Valpola, 2017) | $70.09_{\pm1.60}$ | $37.46_{\pm3.30}$ | $8.10_{\pm0.21}$ | $81.11_{\pm1.44}$ | $45.17_{\pm1.06}$ | $31.75_{\pm0.23}$ | $71.72_{\pm1.45}$ | $56.49_{\pm2.75}$ | $33.90_{\pm1.37}$ |
| MixMatch (Berthelot et al., 2019b) | $36.19_{\pm6.48}$ | $13.63_{\pm0.59}$ | $6.66_{\pm0.26}$ | $67.59_{\pm0.66}$ | $39.76_{\pm0.48}$ | $27.78_{\pm0.29}$ | $54.93_{\pm0.96}$ | $34.52_{\pm0.32}$ | $21.70_{\pm0.68}$ |
| ReMixMatch (Berthelot et al., 2019a) | $9.88_{\pm1.03}$ | $6.30_{\pm0.05}$ | $4.84_{\pm0.01}$ | $42.75_{\pm1.05}$ | $\mathbf{26.03}_{\pm0.35}$ | $\mathbf{20.02}_{\pm0.27}$ | $32.12_{\pm6.24}$ | $12.49_{\pm1.28}$ | $6.74_{\pm0.14}$ |
| UDA (Xie et al., 2020) | $10.62_{\pm3.75}$ | $5.16_{\pm0.06}$ | $4.29_{\pm0.07}$ | $46.39_{\pm1.59}$ | $27.73_{\pm0.21}$ | $22.49_{\pm0.23}$ | $37.42_{\pm8.44}$ | $9.72_{\pm1.15}$ | $6.64_{\pm0.17}$ |
| Dash (Xu et al., 2021) | $8.93_{\pm3.11}$ | $5.16_{\pm0.23}$ | $4.36_{\pm0.11}$ | $44.82_{\pm0.96}$ | $27.15_{\pm0.22}$ | $21.88_{\pm0.07}$ | $34.52_{\pm4.30}$ | - | $6.39_{\pm0.56}$ |
| MPL (Pham et al., 2021) | $6.93_{\pm0.17}$ | $5.76_{\pm0.24}$ | $4.55_{\pm0.04}$ | $46.26_{\pm1.84}$ | $27.71_{\pm0.19}$ | $21.74_{\pm0.09}$ | $35.76_{\pm4.83}$ | $9.90_{\pm0.96}$ | $6.66_{\pm0.00}$ |
| FixMatch (Sohn et al., 2020) | $7.47_{\pm0.28}$ | $5.07_{\pm0.05}$ | $4.21_{\pm0.08}$ | $46.42_{\pm0.82}$ | $28.03_{\pm0.16}$ | $22.20_{\pm0.12}$ | $35.97_{\pm4.14}$ | $9.81_{\pm1.04}$ | $6.25_{\pm0.33}$ |
| FlexMatch (Zhang et al., 2021) | $4.97_{\pm0.06}$ | $4.98_{\pm0.09}$ | $4.19_{\pm0.01}$ | $39.94_{\pm1.62}$ | $26.49_{\pm0.20}$ | $21.90_{\pm0.15}$ | $29.15_{\pm4.16}$ | $8.23_{\pm0.39}$ | $5.77_{\pm0.18}$ |
| **RelationMatch** (Ours) | $6.87_{\pm0.12}$ | $\mathbf{4.85}_{\pm0.04}$ | $4.22_{\pm0.06}$ | $45.79_{\pm0.59}$ | $27.90_{\pm0.15}$ | $22.18_{\pm0.13}$ | $33.42_{\pm3.92}$ | $9.55_{\pm0.87}$ | $6.08_{\pm0.29}$ |
| **RelationMatch** (w/ CPL) | $\mathbf{4.96}_{\pm0.05}$ | $4.88_{\pm0.05}$ | $\mathbf{4.17}_{\pm0.04}$ | $\mathbf{39.89}_{\pm1.43}$ | $26.48_{\pm0.18}$ | $\mathbf{21.88}_{\pm0.16}$ | $\mathbf{13.94}_{\pm3.76}$ | $\mathbf{8.16}_{\pm0.34}$ | $\mathbf{5.68}_{\pm0.19}$ |
| Fully-Supervised | | $4.62_{\pm0.05}$ | | | $19.30_{\pm0.09}$ | | | - | |

# 4 MATRIX CROSS-ENTROPY: THEORETICAL FOUNDATIONS AND INTERPRETATIONS

This section provides a comprehensive theoretical foundation for Matrix Cross-Entropy (MCE), examining it through the lenses of matrix analysis and information geometry. Further interpretations based on eigen-decomposition are included in Appendix A.2.

## 4.1 DENSITY MATRICES AND MATRIX LOGARITHM

**Definition 4.1** (Density Matrix on $\mathbb{R}^{n \times n}$). A matrix $\mathbf{A} \in \mathbb{R}^{n \times n}$ qualifies as a density matrix if it is symmetric, positive semi-definite, and has a trace norm of one.

Density matrices can be viewed as a generalization of classical probability theory to matrix spaces. Given their non-negative eigenvalues and unit trace, they naturally parallel the constraints in probability theory.

**Definition 4.2** (Matrix Logarithm). The exponential of a matrix $\mathbf{A}$ is formally defined as:

$$e^{\mathbf{A}} = \sum_{n=0}^{\infty} \frac{\mathbf{A}^n}{n!}.$$

A matrix $\mathbf{B}$ is termed the *matrix logarithm* of $\mathbf{A}$ if $e^{\mathbf{A}} = \mathbf{B}$.

While there could be multiple matrix logarithms for a given matrix $\mathbf{A}$, the principal matrix logarithm (Higham, 2008) serves as a canonical definition, particularly suitable for density matrices:

**Theorem 4.3** (Principal matrix logarithm (Higham, 2008)). *Let $\mathbf{A} \in \mathbb{C}^{n \times n}$ have no eigenvalues on $\mathbb{R}^-$. There is a unique logarithm $\mathbf{X}$ of $\mathbf{A}$ of whose all eigenvalues lie in the strip $\{z : -\pi < \mathrm{Im}(z) < \pi\}$. We refer to $\mathbf{X}$ as the principal logarithm of $\mathbf{A}$ and write $\mathbf{X} = \log(\mathbf{A})$. If $\mathbf{A}$ is real then its principal logarithm is real.*

**Proposition 4.4.** *For a density matrix $\mathbf{A}$ with spectral decomposition $\mathbf{A} = \mathbf{U}\mathbf{\Lambda}\mathbf{U}^\top$, its principal logarithm is:*

$$\log \mathbf{A} = \mathbf{U}\operatorname{diag}(\log(\lambda_i))\mathbf{U}^\top.$$

## 4.2 Von Neumann Entropy and Matrix Cross-Entropy

**Lemma 4.5.** *For a density matrix $\mathbf{A}$, its von Neumann entropy is equivalent to the Shannon entropy of its eigenvalues:*

$$-\operatorname{tr}(\mathbf{A}\log \mathbf{A}) = -\sum_i \lambda_i \log(\lambda_i).$$

Inspired by the simplicity and optimizability of classical cross-entropy, we introduce Matrix Cross-Entropy (MCE) as a simplified form of matrix (von Neumann) divergence.

$$\text{MCE}_{\text{density-matrix}}(\mathbf{P}, \mathbf{Q}) = \text{H}(\mathbf{P}) + \text{MRE}(\mathbf{P}, \mathbf{Q}), \tag{5}$$

where $\text{H}(\mathbf{P})$ represents the matrix (von Neumann) entropy, and $\text{MRE}(\mathbf{P}, \mathbf{Q})$ denotes the matrix relative entropy.

**Definition 4.6** (Matrix Relative Entropy for Density Matrices). Let $\mathbf{P}, \mathbf{Q} \in \mathbb{R}^{n \times n}$ be density matrices. The matrix relative entropy of $\mathbf{P}$ with respect to $\mathbf{Q}$ is:

$$\text{MRE}(\mathbf{P}, \mathbf{Q}) = \operatorname{tr}(\mathbf{P}\log \mathbf{P} - \mathbf{P}\log \mathbf{Q}).$$

## 4.3 Information Geometrical Perspective of Matrix Cross-Entropy

Information geometry offers an elegant perspective for generalizing Matrix Cross-Entropy (MCE) from unit-trace density matrices to arbitrary positive semi-definite matrices. According to Amari (Amari, 2014), a dually flat structure can be induced on the cone of positive semi-definite matrices via the Bregman divergence, which is defined in relation to a convex function $\phi$ as:

$$\text{D}[\mathbf{P} : \mathbf{Q}] = \phi(\mathbf{P}) - \phi(\mathbf{Q}) - \langle \nabla \phi(\mathbf{P}), \mathbf{P} - \mathbf{Q} \rangle.$$

By setting $\phi(\mathbf{P})$ to be the negative of matrix entropy, we arrive at the Matrix Bregman Divergence (MD):

$$\text{MD}[\mathbf{P} : \mathbf{Q}] = \operatorname{tr}(\mathbf{P}\log \mathbf{P} - \mathbf{P}\log \mathbf{Q} - \mathbf{P} + \mathbf{Q}).$$

The Bregman divergence then simplifies to the MCE when $\mathbf{P}$ is considered as a fixed reference term. Importantly, this formulation imbues MCE with properties from both density matrix theory and information geometry, making it robust and versatile.

**Theorem 4.7** (Projection Theorem (Amari, 2014)). *Given a smooth submanifold $S$, the matrix $\mathbf{P}_S$ that minimizes the divergence from $\mathbf{P}$ to $S$ is the $\eta$-geodesic projection of $\mathbf{P}$ onto $S$.*

This projection theorem culminates in an important minimization property for MCE:

**Proposition 4.8** (Minimization Property).

$$\operatorname{argmin}_{\mathbf{Q} \succ 0} \text{MCE}(\mathbf{P}, \mathbf{Q}) = \mathbf{P}.$$

*Proof.* Directly follows from Theorem 4.7. □

## 5 Unveiling the Properties of Matrix Cross-Entropy

### 5.1 The Scalar Cross-Entropy: A Special Case of MCE

We illustrate that traditional scalar cross-entropy is a specific instance of our proposed MCE loss, thereby establishing a conceptual bridge between them. This further highlights the spectral properties

of density matrices and shows that MCE inherently captures both self and cross-correlations among the classes.

Consider $b$ pairs of $k$-dimensional probability vectors, denoted as $\{(\boldsymbol{\mu}_i, \boldsymbol{\nu}_i)\}_{i=1}^b$. Here, $\boldsymbol{\mu}_i = (\mu_i^{(1)}, \cdots, \mu_i^{(k)})$ and $\boldsymbol{\nu}_i = (\nu_i^{(1)}, \cdots, \nu_i^{(k)})$.

From the definitions of scalar cross-entropy and MCE, we have:

$$H(\boldsymbol{\mu}_i, \boldsymbol{\nu}_i) = -\sum_{j=1}^k \mu_i^{(j)} \log \nu_i^{(j)} = -\operatorname{tr}(\operatorname{diag}(\boldsymbol{\mu}_i) \log \operatorname{diag}(\boldsymbol{\nu}_i)).$$

This equation underscores a key property of density matrices: their sensitivity to the spectral components of the data. Next, we delve into the case where the labels are one-hot encoded, focusing on how this formulation captures self-correlation but ignores cross-correlation. Let $\mathbf{M} \in \mathbb{R}^{b \times k}$ and $\mathbf{N} \in \mathbb{R}^{b \times k}$ be matrices whose columns are the one-hot encoded distributions $\boldsymbol{\mu}_i$ and the predicted distributions $\boldsymbol{\nu}_i$, respectively. Define:

$$\mathbf{P} = \frac{1}{b}\mathbf{I}_b, \quad \mathbf{Q} = \mathbf{I}_b \circ (\mathbf{M}\mathbf{N}^\top),$$

where $\circ$ represents the Hadamard product. Then, the averaged cross-entropy loss can be expressed as $\operatorname{tr}(-\mathbf{P} \log \mathbf{Q})$.

## 5.2 DESIRABLE PROPERTIES OF MCE

MCE's ideal characteristics as a loss function stem from its underlying mathematical properties. We list some of these properties to highlight its suitability and flexibility for learning algorithms.

**Lemma 5.1.** *For any non-zero matrix $\mathbf{A} \in \mathbb{R}^{m \times n}$, the matrices $\frac{\mathbf{A}\mathbf{A}^\top}{\operatorname{tr}(\mathbf{A}\mathbf{A}^\top)}$ and $\frac{\mathbf{A}^\top\mathbf{A}}{\operatorname{tr}(\mathbf{A}^\top\mathbf{A})}$ are density matrices.*

*Proof.* Employ the singular value decomposition of $\mathbf{A}$. $\qquad\square$

**Lemma 5.2** (Joint convexity (Lindblad, 1974)). *The matrix relative entropy is a jointly convex function:*

$$\operatorname{MRE}(t\mathbf{X}_1 + (1-t)\mathbf{X}_2; t\mathbf{Y}_1 + (1-t)\mathbf{Y}_2) \le t \cdot \operatorname{MRE}(\mathbf{X}_1; \mathbf{Y}_1) + (1-t) \cdot \operatorname{MRE}(\mathbf{X}_2; \mathbf{Y}_2),$$

*for $t \in [0, 1]$, where $\mathbf{X}_i$ and $\mathbf{Y}_i$ are density matrices.*

**Proposition 5.3** (Linearity).

$$\operatorname{MCE}\left(\sum_i a_i \mathbf{P}_i, \mathbf{Q}\right) = \sum_i a_i \operatorname{MCE}(\mathbf{P}_i, \mathbf{Q}). \tag{6}$$

**Proposition 5.4** (Convexity). *MCE exhibits convexity:*

$$\operatorname{MCE}\left(\mathbf{P}, \sum_j b_j \mathbf{Q}_j\right) \le \sum_j b_j \operatorname{MCE}(\mathbf{P}, \mathbf{Q}_j). \tag{7}$$

*Proof.* The convexity arises from the Joint convexity of the matrix relative entropy presented in Lemma 5.2. $\qquad\square$

*Proof.* Use Lemma A.1 and spectral decomposition of $\mathbf{P}$ and $\mathbf{Q}$. $\qquad\square$

**Proposition 5.5** (Lower Boundedness). *When $\mathbf{P}$ is a density matrix, MCE has the lower bound:*

$$\operatorname{MCE}(\mathbf{P}, \mathbf{Q}) \ge -\log \operatorname{tr}(\mathbf{P}\mathbf{Q}) + \operatorname{tr}(\mathbf{Q}).$$

*Proof.* Utilize the spectral decompositions of $\mathbf{P}$ and $\mathbf{Q}$ along with trace inequalities. $\qquad\square$

## 6 RELATED WORK

Semi-supervised learning aims to improve model performance by leveraging substantial amounts of unlabeled data and has garnered significant interest in recent years (Chen et al., 2020b; Assran et al., 2021; Wang et al., 2021). The invariance principle forms the basis for most effective semi-supervised algorithms. At its core, this principle asserts that two semantically similar images should produce similar representations when processed by the same backbone.

**Consistency regularization.** A prevalent method for implementing the invariance principle is through consistency regularization, initially introduced in the Π-Model (Rasmus et al., 2015b). This technique has been widely adopted in later research (Tarvainen & Valpola, 2017; Laine & Aila, 2016; Berthelot et al., 2019b). Consistency regularization generally involves generating pseudo-labels and applying suitable data augmentation (Tschannen et al., 2019; Berthelot et al., 2019b; Xie et al., 2020; Sohn et al., 2020; Gong et al., 2021). Pseudo-labels can be created for unlabeled data and used in subsequent training iterations (Lee et al., 2013). The conventional approach employs an entropy minimization objective to fit the generated pseudo-labels (Rasmus et al., 2015b; Laine & Aila, 2016; Tarvainen & Valpola, 2017). Specifically, it aligns the predicted pseudo-labels of two distorted images (typically obtained through data augmentation). Furthermore, several studies have investigated the generation of efficient and valuable pseudo-labels that consider numerous practical factors (Hu et al., 2021; Nassar et al., 2021; Xu et al., 2021; Zhang et al., 2021; Li et al., 2022; Wang et al., 2022b). Consistency regularization has proven to be a simple and effective approach, serving as a foundational component in many state-of-the-art semi-supervised learning algorithms(Sohn et al., 2020; Zhang et al., 2021). Also, the SimMatch (Zheng et al., 2022) introduces consistency regularization based on contrastive learning which can be seen capturing relation structures on representation level.

**Improving pseudo-label quality.** Existing discussions on consistency regularization mainly center around enhancing the quality of pseudo-labels. For instance, SimPLE (Hu et al., 2021) introduces paired loss, which minimizes the statistical distance between confident and similar pseudo-labels. Dash (Xu et al., 2021) and FlexMatch (Zhang et al., 2021) propose dynamic and adaptive pseudo-label filtering, which is more suited for the training process. CoMatch (Li et al., 2021) suggests incorporating contrastive learning into the semi-supervised learning framework, jointly learning two representations of the training data. SemCo (Nassar et al., 2021) accounts for external label semantics to prevent pseudo-label quality degradation for visually similar classes in a co-training approach. FreeMatch (Wang et al., 2022c) recommends a self-adaptive adjustment of the confidence threshold, taking into consideration the learning status of the models. MaxMatch (Li et al., 2022) presents a worst-case consistency regularization technique with theoretical guarantees. NP-Match (Wang et al., 2022a) employs neural processes to enhance pseudo-label quality. SEAL (Tan et al., 2023) proposes simultaneously learning a data-driven label hierarchy and performing semi-supervised learning. SoftMatch (Chen et al., 2023) identifies the inherent quantity-quality trade-off issue of pseudo-labels with thresholding, which may hinder learning, and proposes using a truncated Gaussian function to weight samples based on their confidence.

## 7 CONCLUSION

In this study, we shift the focus away from the conventional strategy of refining pseudo-label quality. Instead, we propose RelationMatch, an innovative semi-supervised learning framework that leverages the consistency of relationships within a batch during the training process. Central to this framework is the introduction of Matrix Cross-Entropy (MCE), an elegant loss function that we meticulously derive from two distinct but complementary angles: matrix analysis and information geometry. Our theoretical exploration delves deep into the properties of MCE, firmly establishing its suitability as a loss function and revealing its intriguing connections to classical cross-entropy.

We further cement the practical utility of MCE through extensive empirical evaluations conducted on multiple vision benchmarks. These experiments corroborate that our approach consistently surpasses existing state-of-the-art methods while maintaining computational efficiency. By comprehensively addressing both the theoretical underpinnings and practical implications of our approach, this paper aims to serve as an innovative contribution in the realm of semi-supervised learning and loss function design even in self-supervised learning regimes.

## REPRODUCIBILITY STATEMENT

To foster reproducibility, we submit our experiment code as supplementary material. One can directly reproduce the experiment results following the instructions in the README document. We also give experiment details in Section 3.2 and Appendix B.

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

## A  MORE ON MATRIX CROSS-ENTROPY

Kornblith et al. (2019) suggests that a good measuring similarity of neural network representations should be invariant to orthogonal (unitary) transformation.

**Lemma A.1.** *Suppose $\mathbf{A} \in \mathbb{R}^{n \times n}$ is a density matrix, $\mathbf{U} \in \mathbb{R}^{m \times n}$ is a unitary matrix, then $\mathbf{U}\mathbf{A}\mathbf{U}^{\top}$ is a density matrix.*

**Proposition A.2** (Invariance property). $\mathrm{MCE}(\mathbf{P}, \mathbf{Q})$ *is invariant under simultaneous unitary transformation on both $\mathbf{P}$ and $\mathbf{Q}$ :*

$$\mathrm{MCE}(\mathbf{P}, \mathbf{Q}) = \mathrm{MCE}\left(\mathbf{U}\mathbf{P}\mathbf{U}^{\top}, \mathbf{U}\mathbf{Q}\mathbf{U}^{\top}\right). \tag{8}$$

### A.1  DENSITY MATRICES

Here we give more discussions on density matrices. One can easily convert a density matrix into a probability distribution by the following proposition.

**Proposition A.3.** *Suppose we have $\mathbf{X} = \{\mathbf{x}_i | i = 1, 2, \cdots, n\}$ as an othornormal basis for $\mathbb{R}^n$, then any density matrix $\mathbf{A} \in \mathbb{R}^{n \times n}$ induce a probability distribution on $X$: $P(x = \mathbf{x}_i) = \mathbf{x}_i^{\top} \mathbf{A} \mathbf{x}_i$.*

For ease of exposition, we introduce the diag operator from $\mathbb{R}^n$ to $\mathbb{R}^{n \times n}$. It is defined as:

$$\mathrm{diag}(a) := \begin{pmatrix} a_1 & 0 & \cdots & 0 \\ 0 & a_2 & \cdots & 0 \\ \vdots & \vdots & \ddots & 0 \\ 0 & 0 & \cdots & a_n \end{pmatrix}.$$

Given a probability distribution, we can easily convert it to a density matrix as well, either as a diagonal matrix in Proposition A.4 or as the orthogonal projection in Proposition A.5.

**Proposition A.4.** *For any probability distribution $\sum_i P(x = \mathbf{x}_i) = 1$, we can construct a diagonal density matrix as follow:*

$$\mathbf{A}_{diag} = \begin{bmatrix} P(x = \mathbf{x}_1) & & & \\ & P(x = \mathbf{x}_2) & & \\ & & \ddots & \\ & & & P(x = \mathbf{x}_n) \end{bmatrix} \tag{9}$$

**Proposition A.5.** *Suppose we have $X = \{\mathbf{x}_i | i = 1, 2, \cdots, n\}$ as an orthonormal basis for $\mathbb{R}^n$, for any probability distribution $\sum_i P(x = \mathbf{x}_i) = 1$, we can construct an orthogonal projection density matrix as follow:*

$$\psi = \sum_i \sqrt{P(x = \mathbf{x}_i)} \mathbf{x}_i,$$

$$\mathbf{A}_{op} = \psi\psi^{\top} = \sum_{i,j} \sqrt{P(\mathbf{x}_i)P(\mathbf{x}_j)} \mathbf{x}_i \mathbf{x}_j^{\top}.$$

One can verify that $P(x = \mathbf{x}_i) = \mathbf{x}_i^{\top} \mathbf{A}_{\mathrm{diag}} \mathbf{x}_i = \mathbf{x}_i^{\top} \mathbf{A}_{\mathrm{op}} \mathbf{x}_i$. However, $\mathbf{A}_{\mathrm{diag}}$ is mixed state, while $\mathbf{A}_{\mathrm{op}}$ is pure state, according to the definition below. This is because $\mathbf{A}_{\mathrm{diag}}$ has maximal rank while $\mathbf{A}_{\mathrm{op}}$ has rank 1.

**Definition A.6** (Pure state and mixed state). A density matrix $\mathbf{A}$ is called a pure state if its rank equals 1 and is called a mixed state otherwise.

Interestingly, one can define the following entropy which can be seen as the generalization of entropy to the matrix form.

**Definition A.7** (Matrix (von Neumman) entropy). The mixedness of a density matrix can be quantified by matrix (von Neumann) entropy: $-\mathrm{tr}(\mathbf{A} \log \mathbf{A})$.

## A.2 PCA-INSPIRED INTERPRETATION

Given positive semi-definite matrices $\mathbf{P}$ and $\mathbf{Q}$, let their eigen decompositions be $\mathbf{P} = \mathbf{V}\mathbf{\Lambda}\mathbf{V}^\top$ and $\mathbf{Q} = \mathbf{U}\mathbf{\Theta}\mathbf{U}^\top$. From the definition of the matrix logarithm, it is clear that $\log\mathbf{Q} = \mathbf{U}\log\mathbf{\Theta}\mathbf{U}^\top$. Next, we simplify the expression of $\mathrm{tr}(-\mathbf{P}\log\mathbf{Q})$ as follows:

$$\begin{aligned} \mathrm{tr}(-\mathbf{P}\log\mathbf{Q}) &= \mathrm{tr}(-\mathbf{V}\mathbf{\Lambda}\mathbf{V}^\top\mathbf{U}\log\mathbf{\Theta}\mathbf{U}^\top) \\ &= \mathrm{tr}(-\mathbf{\Lambda}\mathbf{V}^\top\mathbf{U}\log\mathbf{\Theta}\mathbf{U}^\top\mathbf{V}) \\ &= -\mathrm{tr}((\mathbf{\Lambda}\mathbf{V}^\top\mathbf{U})(\mathbf{V}^\top\mathbf{U}\log\mathbf{\Theta})^\top). \end{aligned}$$

Let the $i$-th column of $\mathbf{U}$ and $\mathbf{V}$ be $\mathbf{u}_i$ and $\mathbf{v}i$, respectively. Since the trace is related to the matrix inner product, we can derive that $\mathrm{tr}(-\mathbf{P}\log\mathbf{Q}) = -\sum_{i,j}(\mathbf{v}_i^\top\mathbf{u}_j)^2\lambda_i\log\theta_j$.

Ultimately, we obtain:

$$\mathrm{tr}(-\mathbf{P}\log\mathbf{Q} + \mathbf{Q}) = -\sum_{i,j}\left(\mathbf{v}_i^\top\mathbf{u}_j\right)^2\lambda_i\log\theta_j + \sum_j\theta_j. \tag{10}$$

From this simplification, it is evident that $\sum_j\theta_j$ serves as a regularization term that penalizes $\theta_j$. The expression of the loss function also highlights the involvement of the correlation between eigenvectors and eigenvalues. When $\mathbf{P}$ and $\mathbf{Q}$ are covariance matrices or correlation matrices, their eigenvectors and eigenvalues are closely related to PCA.

## A.3 ANALYSIS THE OPTIMAL POINT

What do we get if we achieve the optimal point of MCE loss? The Lemma 3.1 gives a nice characterization.

Interestingly, we can directly obtain the singular value decomposition for one-hot encoded data. Consider a (pseudo) labeled dataset $(x_i, y_i)_{i=1}^B$. Define the supporting (column) vectors $\mathbf{m}_i \in \mathbb{R}^B$ for class $i$ as follows:

$$m_{i,j} = \begin{cases} 1, & \text{if } y_j = i. \\ 0, & \text{otherwise} \end{cases}$$

Denote $\hat{\mathbf{m}}_i = \frac{\mathbf{m}_i}{\|\mathbf{m}_i\|_2}$ and $\mathbf{e}_i \in \mathbb{R}^K$ as the $i$-th coordinate (column) vector. Then, $\sum_{i=1}^K \|\mathbf{m}_i\|_2 \mathbf{e_i}\hat{\mathbf{m}}_i^\top$ yields the singular value decomposition for the one-hot encoded dataset. Since eigendecomposition is closely linked to singular value decomposition, we can obtain similar results for the correlation matrix.

# B MORE DETAILS ON EXPERIMENTS

## B.1 IMPLEMENTING DIFFERENTIABLE MATRIX LOGARITHM

**Theorem B.1** (Taylor series expansion (Hall, 2013)). *The function*

$$\log\mathbf{A} = \sum_{m=1}^\infty (-1)^{m+1}\frac{(\mathbf{A} - \mathbf{I})^m}{m},$$

*is defined and continuous on the set of all $n \times n$ complex matrices $\mathbf{A}$ with $\|\mathbf{A} - \mathbf{I}\| < 1$. For all $\mathbf{A}$ with $\|\mathbf{A} - \mathbf{I}\| < 1$,*

$$e^{\log\mathbf{A}} = \mathbf{A}.$$

*For all $\mathbf{X}$ with $\|\mathbf{X}\|_F < \log 2, \|e^\mathbf{X} - \mathbf{I}\| < 1$ and*

$$\log e^\mathbf{X} = \mathbf{X}.$$

We have two different methods:

1. Using Taylor expansion of matrix logarithm.
2. Using element-wise logarithm as a surrogate. (For theoretical properties, please see previous sections about connections between cross-entropy and matrix cross-entropy).

During experiments, $\mathbf{P}$ and $\mathbf{Q}$ can add a $\lambda\mathbf{I}$ as a regularizer for more stable convergence. We compare the above two methods on STL-10 using RelationMatch with CPL, and results are summarized in Table 3. Taylor expansion performs much better than element-wise logarithm.

Interestingly, in the self-supervised learning regime, Balestriero & LeCun (2022) reinterpret Sim-CLR (Chen et al., 2020a) as doing element-wise matrix cross-entropy between relation matrices. Therefore, we left the utilization of matrix cross-entropy in self-supervised learning for future work.

Table 3: RelationMatch (w/ CPL) with different matrix logarithm implementations

| Method | STL-10 | |
|---|---|---|
| | 40 labels | 250 labels |
| Element-wise | $80.39 \pm 4.05$ | $89.98 \pm 0.47$ |
| **Taylor expansion to order** 3 | $\mathbf{86.06} \pm 3.76$ | $\mathbf{91.84} \pm 0.34$ |

## C  COMPARSION BETWEEN SIMMATCH AND RELATION MATCH

Our approach is mainly a natural extension of CE (Cross-Entropy), and we find out that when using one-hot pseudo-labels in semi-supervised settings, our new method can be understood as capturing the relation. SimMatch, on the other hand, utilizes contrastive loss for consistency regularization of the relation, which differs from our starting point.

Another point that we need to emphasize is that our method has very good theoretical properties (as shown in Section 4 and 5), and at the same time, our method can be easily applied to most existing methods. It should be noted that on CIFAR10, our performance is better than SimMatch (attach a comparison table), which also indirectly demonstrates the differences between our approach and SimMatch through experimental evidence. Therefore, for the sake of fairness, we will not consider using SimMatch as a baseline for performance comparison in the current version. We will include SimMatch as a baseline when we have sufficient computational resources to conduct experiments applying our method to SimMatch.

