# OpenReview forum: "RelationMatch: Matching In-batch Relationships for Semi-supervised Learning"
_ICLR.cc/2024/Conference — Submitted to ICLR 2024_

### Official Review · Reviewer_SHZg · 2023-10-25

**Soundness:** 4 excellent
**Presentation:** 3 good
**Contribution:** 4 excellent
**Rating:** 8
**Confidence:** 3

**Summary:**

This paper proposes the matrix cross-entropy (MCE) loss for semi-supervised learning (SSL). In addition to matching the output of the strong augmentation with the pseudo-label of the weak augmentation, they also match the pairwise product of the output of the strong augmentation with that of the weak augmentation. Extensive theoretical analysis on the MCE loss reveals the nice theoretical property of the proposed approach. Experiments on benchmark datasets validate the effectiveness of the proposal.

**Strengths:**

- The proposed MCE loss is novel and interesting. As far as I know, this is the first time it has been applied to the SSL literature.
- The theoretical analysis is very comprehensive and sound. The nice theoretical property can promote further investigation of MCE in the community.
- The empirical performance is very strong since the compared methods are very recent and strong methods in SSL.

**Weaknesses:**

My small comment concerns the details of the writing, especially the notations. There may be some typos or unclear statements.

- In section 2.1, it should read $\log q_1$ instead of $\log q_i$.
- In section 2.1, what's Eq.(2.1)?
- In Eq. 4, is $\tilde{Y}_s$ the model output of weak augmentations? In a line above it is written as $\tilde{Y}$. If they mean the same thing, the notation should be the same.
- In Definition 4.2, it should be $n=0$ instead of $i=0$.

The author should check the notation carefully.

**Questions:**

Since MCE can be simplified with a $l_2$-normalized matrix, what is the loss function used in the experiments? Is it still equation (1) with a non-normalized matrix?

---

> ### Author Response · Authors · 2023-11-21
>
> > W:  My small comment concerns the details of the writing, especially the notations. There may be some typos or unclear statements.
> > - In section 2.1, it should read $\log q_1$ instead of $\log q_i$.
> > - In section 2.1, what's Eq.(2.1)?
> > - In Eq. 4, is $\tilde{Y}_s$ the model output of weak augmentations? In a line above it is written as $\tilde{Y}$. If they mean the same thing, the notation should be the same.
> > - In Definition 4.2, it should be n=0 instead of i=0.
> > The author should check the notation carefully.
>
> Thank you for your careful reading and suggestions; we have resolved these issues in the new submission.
>
>
> > Q: Since MCE can be simplified with a $l_2$-normalized matrix, what is the loss function used in the experiments? Is it still equation (1) with a non-normalized matrix?
>
> A: Thank you for your questions. In the experiment, in order to prevent errors caused by some unknown mistakes, we used equation(1) + row l2-normalized $X$ (recall $Q$ = $\frac{1}{b} X X^T$), which makes equation(2) equivalent to equation(1). However, for the sake of caution, we still used equation(1) as our loss in the experiments.

---

> > ### Comment · Reviewer_SHZg · 2023-11-21
> >
> > Thanks for the reply. I would keep my score.

---

### Official Review · Reviewer_Hi55 · 2023-10-31

**Soundness:** 2 fair
**Presentation:** 2 fair
**Contribution:** 1 poor
**Rating:** 3
**Confidence:** 5

**Summary:**

This paper investigates the challenges in semi-supervised learning. The authors highlight that prior research has often overlooked the interconnections between data points in a batch. To address this gap, they introduce RelationMatch, an approach designed to harness the consistency of relationships within a batch of unlabeled data.

**Strengths:**

1. The paper is well written and easy to understand.
2. The authors present the derivation of the proposed MCE Loss through the lenses of matrix analysis and information geometry, showcasing its advantageous characteristics such as convexity, boundedness from below, and optimizable properties.

**Weaknesses:**

1. My primary concern pertains to the paper's novelty. SimMatch[1] has previously addressed the relationship between data points by applying consistency regularization at both the semantic and instance levels, promoting identical class predictions and maintaining similarity relations with other instances for different augmentations of the same instance. A detailed discussion and comparison between SimMatch and RelationMatch are essential to elucidate the distinct contributions of the latter.

2. The benchmark comparison appears outdated. The most recent method evaluated in the paper is from 2021, and although the authors mention some methods from 2022 and 2023, such as FreeMatch, MaxMatch, and NP-Match, these have not been included in the experimental comparisons. When compared with the latest methods, RelationMatch does not seem to meet the state-of-the-art standard.

3. The experimental scope of the paper is limited to toy datasets. To bolster the findings, it is recommended to extend the experiments to more complex, real-world datasets, such as ImageNet.


[1] Zheng, Mingkai, et al. "Simmatch: Semi-supervised learning with similarity matching." Proceedings of the IEEE/CVF Conference on Computer Vision and Pattern Recognition. 2022.

**Questions:**

See weaknesses.

---

> ### Author Response · Authors · 2023-11-21
>
> Thank you for your careful reading and questions.
>
> > W1: My primary concern pertains to the paper's novelty. SimMatch[1] has previously addressed the relationship between data points by applying consistency regularization at both the semantic and instance levels, promoting identical class predictions and maintaining similarity relations with other instances for different augmentations of the same instance. A detailed discussion and comparison between SimMatch and RelationMatch are essential to elucidate the distinct contributions of the latter.
>
> - Our approach is mainly a natural extension of CE (Cross-Entropy), and we find out that when using one-hot pseudo-labels in semi-supervised settings, our new method can be understood as capturing the relation. SimMatch, on the other hand, utilizes contrastive loss for consistency regularization of the relation, which differs from our starting point.
>
> - One point that we need to emphasize is that our method has very good theoretical properties (as shown in Section ...), and at the same time, our method can be easily applied to most existing methods. Therefore, for the sake of fairness, we will not consider using SimMatch as a baseline for performance comparison in the current version. We will include SimMatch as a baseline when we have sufficient computational resources to conduct experiments applying our method to SimMatch.
>
>   We have added these two discussions in the appendix (Section C).
>
> ---
>
> > W2: The benchmark comparison appears outdated. The most recent method evaluated in the paper is from 2021, and although the authors mention some methods from 2022 and 2023, such as FreeMatch, MaxMatch, and NP-Match, these have not been included in the experimental comparisons. When compared with the latest methods, RelationMatch does not seem to meet the state-of-the-art standard.
>
>  Thank you for your careful reading and suggestions. Due to the shortage of computational resources, we only had time to conduct complete experiments on FixMatch / FlexMatch + MCE.
>
>  During the rebuttal period, we attempted to apply our method to FreeMatch. Due to the scarcity of computational resources, we are only able to finish experiments on CIFAR10 and STL-10. The results are shown in the following table, and it can be seen that our method has a significant improvement on the task of CIFAR10 with 40 labels.
>
> | Metric | CIFAR10 40 | CIFAR10 250 | CIFAR10 4000 | STL-10 40 | STL-10 250 | STL-10 1000 |
> |--------|------------|-------------|----------------|-----------|------------|-------------|
> | *FreeMatch* | 4.90 ±0.04 | 4.88 ±0.18 | 4.10 ±0.02 | 15.56 ±0.55 | - | 5.63 ±0.15 |
> | *RelationMatch (w/ self-adaptive threshold)* | 4.80 ±0.06 | 4.89 ±0.14 | 4.08 ±0.05 | 15.60 ±4.09 | 10.42 ±0.78 | 5.62 ±0.18 |
>
>
> What we need to point out is that our core idea is to propose a new loss, MCE, which is a universal enhancement that can be applied to most semi-supervised learning algorithms. We will supplement the experimental results with more advanced methods + MCE as soon as we have sufficient computing resources. Our current implementation of FixMatch / FreeMatch + MCE, when compared to the recent FreeMatch, MaxMatch, and NP-Match, is still not entirely fair.
>
> ---
>
> > W3: The experimental scope of the paper is limited to toy datasets. To bolster the findings, it is recommended to extend the experiments to more complex, real-world datasets, such as ImageNet.
>
> Thank you for your suggestion. We will carry out the ImageNet-related experiments as soon as we have sufficient computing resources available. However, we would like to emphasize that our method performs very well on our current benchmarks, especially when labels are scarce. For example, there is a 15.21% improvement on STL with 40 labels.

---

> ### Author Response · Authors · 2023-11-23
> **Seeking Your Input on Revised Paper's Alignment with ICLR Standards**
>
> Dear Reviewer Hi55,
>
> As the discussion period approaches its conclusion, **we want to ensure that we have thoroughly addressed all your concerns and that our revised paper fully meets the standards of ICLR**. We would highly value any additional feedback you may provide.
>
> Thank you sincerely for your time and consideration.
>
> Best regards,
>
> The Authors

---

> > ### Comment · Reviewer_Hi55 · 2023-11-23
> >
> > Thanks for your reply. I will keep my original score because (1)  experimental comparison with SimMatch is necessarily needed (2) when compared with freematch, the improvement is very marginal (in fact, on many datasets, freematch is better).

---

### Official Review · Reviewer_8nMg · 2023-10-31

**Soundness:** 3 good
**Presentation:** 3 good
**Contribution:** 3 good
**Rating:** 5
**Confidence:** 4

**Summary:**

The paper introduces the consistency between each pair of weak and strong augmentation within a batch in semi-supervised learning.

**Strengths:**

1. the paper proposes a novel idea, which consider in-batch relationship in SSL.
2. The paper proposes matrix cross-entropy, which has a theoretical foundation and interpretations.
3. Good writing, easy to follow, I appreciate the warm-up example, which is helpful for understanding.

**Weaknesses:**

1. Figure 1 can be improved. There are too many lines, which are confusing.
2. Large dataset experiments are missing, e.g., ImageNet
3. Ablation studies on $\mu$ and $\gamma$ are missing.
4. Formulations and notations are not clear. What's the definition of $Y_s$ and $X_s$ in eq(4)?
5. How MCE connect with Relation in the introduction?

**Questions:**

1. Would you consider a relation (strongeaug dog, strongaug cat) > relation(weakaug dog, weakaug cat)? Intuitively, this relation more close to nature's rule.
2. Is there an intuitive explanation of matrix cross-extropy?
3. MCE(P, Q) = tr(−P log Q + Q). For matrix cross-entropy, why +Q?

---

> ### Author Response · Authors · 2023-11-21
>
> Thank you for your careful reading and questions.
>
> > W1 : Figure 1 can be improved. There are too many lines, which are confusing.
>
> Thank you for your suggestion, we will fix this issue in the next submission. We deeply apologize for any confusion caused to you.
>
> > W2: Large dataset experiments are missing, e.g., ImageNet
>
> Thank you for your suggestion. We will carry out the ImageNet-related experiments as soon as we have enough computing resources. However, we would like to emphasize that our method has performed very well on our existing benchmarks, especially when labels are scarce. For instance, there is an improvement of 15.21 on STL with 40 labels, an increase of 0.6 (FixMatch) on CIFAR10 with 40 labels, and a rise of 0.63 (FixMatch) on CIFAR100 with 40 labels.
>
> > W3: Ablation studies on $\mu$ and $\gamma$ are missing.
>
> Thank you for your suggestion. Please note that we have already emphasized in the original text that our hyperparameter selection is made to ensure that the coefficient of the cross-entropy loss in the loss function is consistent with that of the baseline, in order to make a fair comparison.
>
> > W4: Formulations and notations are not clear. What's the definition of $Y_s$ and $X_s$ in eq(4)?
>
> Thank you for your question. We have made changes to the manuscript to make the equation clearer.
>
> > W5: How MCE connect with Relation in the introduction?
>
> Thank you for your question. We have discussed in Section 2.1 that the relation matrix can be easily generated by the predicted probability. Note MCE has minimization property(Proposition 4.8), so it can match the relation matrix of weakly and strongly augmented images from RelationMatch loss given in equation (4).
>
> > Q1: Would you consider a relation (strongeaug dog, strongaug cat) > relation(weakaug dog, weakaug cat)? Intuitively, this relation more close to nature's rule.
>
> Thank you for your question. In this paper, we consider improving SSL by incorporating matching the total relationship of **a batch** of weakly augmented images to their strongly augmented counterpart batches. This matching forces consistency regularization, not an unequal relationship like the one you show. We only consider consistency regularization mainly as this philosophy is shown to be beneficial in semi-supervised learning among literatures.
>
> > Q2: Is there an intuitive explanation of matrix cross-extropy?
>
> Yes. When P and Q have all its trace 1 (density matrix), the MCE can be seen as using cross-entropy on the spectrum of P and Q.
>
> > Q3: MCE(P, Q) = tr(−P log Q + Q). For matrix cross-entropy, why +Q?
>
> This is for better theoretical properties for matrix Q when it is a positive semi-definite matrix and not a matrix whose trace is 1, we generalize the probability distribution defined in the density matrix (the standard result used in quantum information theory) into any positive semi-definite matrix, with multiple theoretical guarantees.

---

> ### Author Response · Authors · 2023-11-23
> **Seeking Your Input on Revised Paper's Alignment with ICLR Standards**
>
> Dear Reviewer 8nMg,
>
> As the discussion period approaches its conclusion, **we want to ensure that we have thoroughly addressed all your concerns and that our revised paper fully meets the standards of ICLR**. We would highly value any additional feedback you may provide.
>
> Thank you sincerely for your time and consideration.
>
> Best regards,
>
> The Authors

---

### Official Review · Reviewer_BTXg · 2023-11-03

**Soundness:** 2 fair
**Presentation:** 2 fair
**Contribution:** 3 good
**Rating:** 5
**Confidence:** 3

**Summary:**

This paper studies the problem of semi-supervised learning, which is a common and interesting area. The author proposes RelationMatch, an innovative semi-supervised learning framework that capitalizes on these relationships through a novel Matrix Cross-Entropy (MCE) loss function. Extensive empirical evaluations, including a 15.21% accuracy improvement over FlexMatch on the STL-10 dataset, have demonstrated that RelationMatch consistently outperforms existing state-of-the-art methods.corruptions.

**Strengths:**

1. This paper is well-written, well-organized, and easy to follow.
2. The paper addresses a novel and important problem, i.e., the relationships among data points within a batch, which has not been well-studied in the literature.
3. This method can be easily incorporated with other works

**Weaknesses:**

1. The experiment appears somewhat insufficient, as only two experiments were conducted in the main text, and they were tested on just two to three datasets. Additionally, I am curious as to why the STL-10 dataset was omitted from Table 1.
2. Based on the results presented in Table 1, the displayed accuracy results show limited differentiation. The matrix cross-entropy outperformed by a margin of less than 0.3%. This could potentially be attributed to randomization and perturbations.
3. Potential failure modes or limitations not discussed.

**Questions:**

The primary questions for the rebuttal primarily arise from the "weaknesses" section.

---

> ### Author Response · Authors · 2023-11-21
>
> > W1: The experiment appears somewhat insufficient, as only two experiments were conducted in the main text, and they were tested on just two to three datasets. Additionally, I am curious as to why the STL-10 dataset was omitted from Table 1.
>
> Thank you for your suggestions; we will add more experiments in the future version. As for the omission of the STL-10 dataset in Table 1, this is mainly because STL-10 essentially is not suited for supervised training, since the majority of images it contains are unlabeled, and within the labeled images, each class has 800 test images and 500 train images, which is clearly an unbalanced ratio for conducting supervised experiments.
>
> > W2: Based on the results presented in Table 1, the displayed accuracy results show limited differentiation. The matrix cross-entropy outperformed by a margin of less than 0.3%. This could potentially be attributed to randomization and perturbations.
>
> Thank you for your careful reading and questions. In order to minimize the impact of randomness, we have repeated the experiment **3** times and reported the means and variances. Additionally, we must point out that, in supervised learning, when the baseline accuracy has already surpassed 95%, even an improvement of 0.3% is very significant.
>
> >W3: Potential failure modes or limitations not discussed.
>
> Thank you for your suggestion. We will try to incorporate potential failure modes as well as limitations after a thorough investigation.

---

> ### Author Response · Authors · 2023-11-23
> **Seeking Your Input on Revised Paper's Alignment with ICLR Standards**
>
> Dear Reviewer BTXg,
>
> As the discussion period approaches its conclusion, **we want to ensure that we have thoroughly addressed all your concerns and that our revised paper fully meets the standards of ICLR**. We would highly value any additional feedback you may provide.
>
> Thank you sincerely for your time and consideration.
>
> Best regards,
>
> The Authors

---

### Author Response · Authors · 2023-11-21
**General response**

Thank all the reviewers for their suggestions and questions. In line with the reviewers' suggestions, we have made minor revisions, including optimizing some of the notation in Eq. 4, improving Figure 1, and fixing typos mentioned by the reviewers. We have also referenced Simmatch[1] and included a comparative discussion with Simmatch.

[1] Zheng, Mingkai, et al. "Simmatch: Semi-supervised learning with similarity matching." Proceedings of the IEEE/CVF Conference on Computer Vision and Pattern Recognition. 2022.

---

### Meta-Review · Area_Chair_hVby · 2023-12-14

**Metareview:**

This paper introduces RelationMatch, an innovative semi-supervised learning framework addressing the overlooked relationships among data points within a batch. Leveraging a novel Matrix Cross-Entropy (MCE) loss function derived from matrix analysis and information geometry, RelationMatch consistently outperforms existing state-of-the-art methods. Extensive empirical evaluations, including a notable 15.21% accuracy improvement over FlexMatch on the STL-10 dataset, validate its effectiveness. The proposed Matrix Cross-Entropy (MCE) Loss, derived through matrix analysis and information geometry, exhibits favorable characteristics and theoretical properties, making it a noteworthy contribution to semi-supervised learning literature. The method's applicability to other works and its strong empirical performance further enhance its significance in the field. While the paper is well-organized, concerns arise regarding the experiment's scope, with limited datasets and minimal differentiation in accuracy results. The omission of the STL-10 dataset and potential failure modes are notable gaps. The novelty of RelationMatch is questioned, urging a detailed comparison with SimMatch. The benchmark comparison is deemed outdated, lacking evaluation against recent methods like FreeMatch and MaxMatch. Furthermore, the experimental focus on toy datasets is flagged, suggesting an extension to real-world datasets for a more comprehensive assessment. After the rebuttal, most of the reviewers' concerns have not been adequately addressed. Therefore, I have determined that this paper does not meet the acceptance criteria and recommend rejecting it.

**Justification For Why Not Higher Score:**

This paper lacks innovation and rigor, with predominantly negative feedback from reviewers and significant disagreements. Therefore, it is not suitable for a recommendation for acceptance.

**Justification For Why Not Lower Score:**

N/A

---

### Decision · Program_Chairs · 2024-01-16

Reject